# Effectiveness of a Two-Year Multicomponent Intervention for the Treatment of Overweight and Obesity in Older People

**DOI:** 10.3390/nu14224762

**Published:** 2022-11-11

**Authors:** Lorena Rumbo-Rodríguez, Ana Zaragoza-Martí, Miriam Sánchez-SanSegundo, Rosario Ferrer-Cascales, Ana Laguna-Pérez, Jose A. Hurtado-Sánchez

**Affiliations:** 1Health Psychology and Human Behavior Research Group, Nursing Department, Faculty of Health Sciences, University of Alicante, 03690 Alicante, Spain; 2Alicante Institute for Health and Biomedical Research (ISABIAL-FISABIO Foundation), 03010 Alicante, Spain; 3Health Psychology and Human Behavior Research Group, Department of Health Psychology, Faculty of Health Sciences, University of Alicante, 03690 Alicante, Spain

**Keywords:** overweight, obesity, Mediterranean diet, psychological well-being, older people

## Abstract

This study aimed to assess the effectiveness of a two-year intervention based on the Mediterranean diet for the treatment of overweight and obesity in a sample of 51 older people from the Mediterranean city of Alicante (Spain). We also examined the effects of the intervention on psychological well-being. The participants were randomly assigned to the experimental and control groups. The experimental group received group nutritional education sessions, an individualized dietary–nutritional treatment based on a Mediterranean diet, and a physical activity program; the control group received Mediterranean nutritional education in a written format. The experimental group showed a greater loss in weight (*p* = 0.017) and percentage of fat mass (*p* = 0.049), and a greater reduction in body mass index (BMI) (*p* = 0.014) and waist circumference (*p* = 0.010). Both groups improved their depression scores using the PHQ-9; however, no significant improvement was seen in adherence to the Mediterranean diet (PREDIMED) and anxiety level (GAD-7). These results suggest that a two-year intervention based on the Mediterranean diet allows an older population with overweight or obesity to achieve greater weight loss and a greater decrease in BMI, waist circumference, and fat mass percentage. In relation to psychological well-being, depression levels improved at the end of said intervention.

## 1. Introduction

The World Health Organization defines overweight and obesity as an abnormal or excessive accumulation of fat, which represents a risk to health [1]. In Spain, according to the National Health Survey, the prevalence of obesity in adults (≥18 years of age) increased from 7.4 to 17.4% between 1987 and 2017. Obesity is more frequently represented in men than in women, except from 65 years of age; after this age, women outnumbered men (26.32 and 24.71%, respectively). Moreover, considering obesity and overweight as a whole, more than a half of adults have an excess of weight (54.5%) [2].

Processes such as industrialization, urbanization, economic development, and food market globalization have contributed to changes in the eating habits of the population. These changes are characterized by an increased choice of food with poor nutritional quality, such as refined, processed, and high-fat foods; more sedentary lifestyles [3] that contribute to overweight and obesity; and the appearance of chronic diseases such as high blood pressure, type II diabetes, cancer, and the development of neurodegenerative diseases [4,5].

The increased prevalence of obesity and overweight is particularly significant in older people, in a period in which the world population is aging due to the decline in birth rates and the progressive increase in life expectancy [6]. In Spain, according to the National Institute of Statistics, on 1 January 2020, the population over 64 years of age was 9.28 million people; between 2002 and 2020, the population aged 64 years increased by more than two million inhabitants [7]. This demographic transition implies that overweight and obesity will have important relevance as they concur with the etiopathogenesis of multiple chronic diseases, such as cardiovascular diseases, type II diabetes, hypertension, and dyslipidemia, whose prevalence increases with age [8,9,10].

Although there is evidence suggesting that interventions based on lifestyle (diet and physical exercise) are successful at promoting weight loss and fat mass in older adults, they remain controversial owing to the associated adverse consequences in nutritional, muscular, and bone health [11,12]. These interventions could lead to an inadequate intake of essential nutrients, such as vitamin D, iron, and calcium [11,13], leading to deficits that could carry an additional associated risk of malnutrition and frailty, with a consequent increased risk of morbidity and mortality [14,15]. Older people are also characterized by important changes in body composition associated with the aging process; lean mass begins to decrease and, consequently, fat mass increases [16]. This loss of muscle mass may lead to the appearance of sarcopenia (alteration of the quantity and quality of muscle mass), which may be aggravated by weight loss, as weight loss implies loss of not only fat mass (FM) but also fat-free mass (FFM); approximately a quarter of all weight lost during these interventions corresponds to FFM [17]. In addition, the loss of mass and strength has been associated with reductions in physical function, quality of life, and frailty, which contribute to the loss of bone density, osteoporosis, and an increased risk of fragility fractures [13,16,18].

Previous studies have reported that intentional, structured, and supervised weight loss interventions are safe and effective for older people to achieve weight loss, leading to positive health benefits [11,13]. For example, Felix and West [13] suggested that behavioral interventions may be effective in achieving significant weight loss without risks for obese older people. Another study observed that interventions that combined dietary–nutritional and physical activity interventions achieved a 10% weight loss in 3 to 12 months, leading to positive changes in physical function, cardiovascular risks, and metabolic results [19]. Finally, Batsis et al. [18] showed that groups with a dietary component in the intervention had a significantly greater weight loss than the groups that only had a physical component. In the same study, dietary interventions were found to be associated with weight loss and improved physical function, whereas interventions based on physical exercise only led to better physical function.

Thus, this study aimed to evaluate the effectiveness of a two-year intervention program based on Mediterranean diet (MD) patterns for the treatment of overweight and obesity in older people. We also examined the effect of the intervention on psychological well-being, given that previous studies have demonstrated that this dietary pattern is a potential means of reducing the risk of depression and anxiety [20,21,22].

## 2. Materials and Methods

### 2.1. Sample and Procedure

The sample included 51 people aged over 60 years (mean = 65.04; standard deviation (SD) = 4.64) from the Mediterranean city of Alicante (Spain). The inclusion criteria were (1) being over 60 years of age and having attended a scheduled consultation with a health professional at the health center and (2) having a body mass index (BMI) greater than 25 kg/m^2^. The exclusion criteria were as follows: (1) patients with a score of three or more errors in the Pfeiffer test (if they had studies) and four or more errors (without studies), (2) having reported difficulties in reading and writing, and (3) having undergone a dietary–nutritional treatment supervised by a nutritionist during the last year. Participants were randomly assigned to two groups: the experimental group and the control group.

Calculations of the ANCOVA global effect at 85% power and an alpha level of 0.05 suggested that 26 participants per group will be needed in order to detect this effect with a medium effect size of 0.25.

From the initial sample, 10 participants were excluded due to inclusion/exclusion criteria. Five participants were excluded because they had followed a dietary treatment over the past year, two participants were excluded because they scored above three errors in the Pfeiffer test, and the other three declined participation. The remaining 51 participants were assigned to the experimental group (n = 25; 88% women, 12% men) or control group (n = 26; 92% women, 8% men). Participants in both groups were matched at baseline for their clinical variables.

The study was approved by the Ethics Committee of the Instituto de Investigación Sanitaria y Biomédica de Alicante (ISABIAL (Health and Biomedical Research Institute of Alicante)) (CEIm: PI2019/057). After being informed regarding the study and the withdrawal process, the participants were requested to participate. The participants were also informed that their participation in the study was voluntary and that they could withdraw from the study without any consequences. Consent was obtained from the entire sample. For those who did not return the signed form, the project staff made several additional attempts.

### 2.2. Sample Selection

Participants were recruited from the Department of Health Alicante, General Hospital (Spain), and were randomly assigned to the control or experimental group, which received 10 group nutritional education sessions and 17 individual sessions with the following objectives: (1) improve inappropriate eating habits, (2) improve nutritional and health status, (3) promote balanced eating habits and healthy lifestyles, and (4) teach strategies for planning healthy menus with appropriate culinary techniques. In addition, each participant received a weight loss history, self-monitoring diary, food calorie book, and visual atlas of portion sizes. The contents of the intervention were prepared and designed by the multidisciplinary research team: dietitians and nutritionists, doctors, psychology staff, nursing staff, and physiotherapists. The design by the research team was based on scientific evidence in the literature and on the recommendations of official bodies such as the World Health Organization.

The multicomponent intervention was carried out in person in the “Health Classroom” of the University of Alicante. As described in Zaragoza-Martí et al. [23], participants in the experimental group received personalized training for weight management, food education, and psychological support and self-care recommendations. The dietary–nutritional intervention was carried out by a trained and qualified dietician–nutritionist. During the individual sessions, a menu adapted to patient needs based on the MD was provided to each participant. In addition, in the follow up individual sessions, the dietitian–nutritionist monitored weight loss by taking anthropometric measurements and evaluated eating behavior by considering questionnaires. During the group sessions, topics related to the MD and its health benefits, the nutritional labeling of packaged foods, the key diet for healthy aging, seasonal diet, the possibility of a healthy gastronomy, diet and cholesterol, diet and hypertension prevention, diet and bone health, healthy snacks and healthy recipes were carried out. In the psychological support sessions, issues related to self-control, anxiety management, emotional feeding, achievements, and difficulties encountered during treatment, review of weekly action plans, relaxation techniques, consolidation of the new image, cognitive distortions, change management, coping with uncertainty, and motivation to change were discussed. The psychological follow-up was carried out by the psychology staff and the follow-up of self-care in health was carried out by the nursing staff. For physical activity, following the recommendations of the World Health Organization (WHO), exercise of at least 150 min per week was recommended.

The control group received nutritional education in a written format on topics such as MD and its benefits, purchasing choices, nutritional food labelling, meal preparation, and healthy culinary techniques.

The intervention program was conducted from September 2019 to October 2021. Every participant was evaluated at baseline and at 6, 12, and 24 months after the baseline evaluation. The data that appear in the following study refer to the baseline evaluations, one-year follow-up evaluations, and final evaluations of each participant.

### 2.3. Eating Habits

#### 2.3.1. Mediterranean Diet Adherence

To determine the degree of adherence to the MD, we used a specific short 14-item questionnaire validated for the Spanish population and used by the Mediterranean Diet Prevention Group (PREDIMED) [24]. To obtain the score, a value of +1 was assigned to each item with a positive connotation for the MD and a value of −1 when the items had a negative connotation. From the sum of the values obtained in the 14 items, the degree of adherence was determined, establishing two levels; if the total score was ≥9, the score showed a good level of adherence, and if the total sum was <9, the score reflected a low level of adherence. To determine participants’ eating habits, a dietary interview was conducted following the guidelines of the Spanish Society of Dietitians–Nutritionists [25].

#### 2.3.2. Food Frequency Questionnaire (MEDIS-FFQ)

The Food Frequency (MEDIS-FFQ) questionnaire was used to assess food consumption, which was validated for the older population living in the Mediterranean [3]. It is composed of 11 food groups, namely: dairy products, cereals and starchy foods, meat, fish, legumes and traditional dishes, vegetables, fruits and nuts, snacks, sweets and salty snacks, beverages, and fats. The frequency of food consumption was assessed in the following categories: daily consumption (once a day or more than twice a day), weekly (from once or twice a week to three to six times a week), monthly (from one to three times a month), and no consumption or occasional consumption [26].

### 2.4. Psychological Well-Being

#### 2.4.1. Patient Health Questionnaire (PHQ-9)

The PHQ-9 (Patient Health Questionnaire) is a self-report tool used to assess the presence and severity of depressive symptoms [27]. It consists of nine Likert-type response items, scored as 0 (never), 1 (several days), 2 (more than half the days), and 3 (almost every day). The total score of the questionnaire ranges from 0 to 27, and the cut-off points of 5, 10, 15, and 20 allow the severity of the symptoms to be classified as follows: 0–4 (minimal), 5–9 (mild), 10–14 (moderate), 15–19 (moderate to severe), and 20–27 (severe) [27,28]. Additionally, the PHQ-9 can be used as a diagnostic algorithm to make a probable diagnosis of major depressive disorder (MDD). MDD should be considered in patients who have had (1) ≥ 5 of the nine depressive symptoms on at least “more than half the days” and (2) one of the first two symptoms (depressed mood or loss of interest). The answers to the questions were based on how the participants felt during the previous two weeks [29,30].

#### 2.4.2. Generalized Anxiety Disorder Scale (GAD-7)

The GAD-7 (generalized anxiety disorder) is a self-report tool used to assess the presence of symptoms of generalized anxiety disorder. It consists of seven Likert-type response items, scored as 0 (never), 1 (several days), 2 (more than half the days), and 3 (almost every day). The total score of this questionnaire ranges from 0 to 21, and the cut-off points of 5, 10, and 15 allow anxiety to be classified as none/normal (0–4), mild (5–9), moderate (10–14), and severe (15–21). The answers to the questions were based on how the participants felt during the previous two weeks [31].

### 2.5. Sociodemographic and Lifestyle Variables

An ad hoc questionnaire was used to collect the sociodemographic, clinical, and lifestyle data. The sociodemographic data included in our study were age, sex, marital status, years of schooling, members of the household, and employment status. The lifestyle variables studied were alcohol and tobacco consumption.

### 2.6. Anthropometric Variables

Standardized methods were used to measure the anthropometric data. Body weight was measured using a vertical mechanical scale with SECA700^®^ sliding weights, with an accuracy of 100 g. Height was measured with an accuracy of 0.2 cm, using the vertical statimeter. Using the data of weight in kg and height in cm, the body mass index (BMI = weight/height^2^; kg/m^2^) was calculated. BMI was interpreted using the World Health Organization classification (BMI < 18.8 = underweight, BMI between 18.5 and 24.99 = normal weight, BMI between 25 and 29.9 = overweight, and BMI ≥ 30 = obesity). Body perimeters were measured in triplicate (after obtaining the mean) using an extendable tape measure. Waist perimeter measurements were performed below the rib cage and above the navel (narrowest waist circumference). The perimeter of the hip was measured horizontally at maximum extension of the buttocks (larger posterior protrusion). The waist–hip index was calculated using the results of both of the measurements. To assess the presence of cardiovascular risk, classification was made according to the waist-hip index (ICC = waist/hip index). In the case of women, the cardiovascular risk was determined when the index was ≥0.85 and, in the case of men, when it was ≥0.94. A clinically validated digital scale with OMRON model HBF-212EW impedance was used to obtain the percentage of fat mass. To assess the percentage of fat mass, the parameters established by the Spanish Society for the Study of Obesity were taken into account; the normal percentage of fat ranges between 12 and 20% in men and between 20 and 30% in women [32].

### 2.7. Statistical Analysis

Descriptive analyses were used to describe the participant characteristics. The normality of the data was confirmed using the Shapiro–Wilk W-test. To assess the intervention effect before, during, and after the intervention, one-way repeated measures ANCOVA was used. For variables that did not meet the normality assumption, the non-parametric Friedman test was used. Variables that presented significant differences (age, level of studies, and employment status) were introduced into the analyses as covariates.

These analyses were carried out using SPSS IBM Corp. (released 2016; IBM SPSS Statistics for Windows, Version 25.0. Armonk, NY: IBM Corp), and the level of statistical significance established for all tests was 0.05.

## 3. Results

### 3.1. Sociodemographic Characteristics

The study sample consisted of 51 individuals assigned to the control group (n = 26) and the experimental group (n = 25). The mean age of the sample was 65.04 years (SD = 4.64), of whom 88.2% were women and the remaining 11.8% were men. Table 1 shows the sociodemographic and lifestyle variables of the participants.

Participants’ mean weight at the beginning of the study was 79.09 kg for the control group, with 65.4% being obese and the remaining 34.6% being overweight. The mean weight for the experimental group at baseline was 82.66 kg, with 80 and 20% of participants being obese and overweight, respectively (Table 2).

The fat mass percentage was high for both groups, exceeding 45%. In reference to cardiovascular risk, the majority of participants in both groups showed cardiovascular risk (control group: 65.3% and experimental group: 60.0%) (Table 2).

### 3.2. Changes in Weight, Body Mass Index, and Percentage of Fat Mass over Time

There was a significant change in weight over time (time F = 3.685; *p* = 0.039), which differed significantly between the control and experimental groups (group × time, F = 4.793; *p* = 0.017), with less weight loss in the control group (Table 3). After 12 and 24 months, weight loss was 2.8 and 2.6 kg greater in the experimental group, respectively (F = 7.049; *p* = 0.011 and F = 3.902; *p* = 0.044, respectively). However, when comparing weight measured at 12 and 24 months, no significant differences were found (F = 1.267); *p* = 0.266).

Regarding BMI, similar to weight, a significant change was observed over time (time F = 3.701; *p* = 0.039), and this change differed significantly between the control and experimental groups (group × time F = 5.020; *p* = 0.014) (Table 3). When comparing the initial BMI and the BMI at 12 and 24 months between groups, significant differences were found (F = 7.460; *p* = 0.009 and F = 4.050; *p* = 0.049, respectively). However, as well as weight, no significant differences were found between BMI measured at 12 months and the one measured at 24 months (F = 1.312; *p* = 0.258).

Regarding the percentage of fat mass, no significant change was observed over time (time F = 2.414; *p* = 0.098). However, there were significant differences in the fat mass percentage between the control and experimental groups over time (group × time F = 3.168; *p* = 0.049). After 12 months, significant differences were found as the percentage of fat mass increased by 0.9% in the control group and decreased by 1.5% in the experimental group (F = 7.117; *p* = 0.012). However, no significant differences were found between groups when comparing fat mass percentage at baseline and at 24 months (F = 1.756; *p* = 0.195), nor when comparing between fat mass percentage at 12 and 24 months (F = 1.193; *p* = 0.283).

### 3.3. Changes in Waist Circumference, Hip Circumference, Waist–Hip Index, and Fat Mass Percentage over Time

As shown in Table 4, there was no significant change in waist circumference over time (time F = 1.430; *p* = 0.245). However, there were significant differences in waist circumference between the control and experimental groups over time (group × time F = 5.248; *p* = 0.010). After 12 months, the waist circumference of the experimental group decreased by an average of 2.3 cm, whereas that of the control group increased by 1.5 cm, leading to significant differences between groups (F = 8.228; *p* = 0.006). Similarly, after 24 months, the waist circumference of the experimental group decreased by an average of 2.5 cm, whereas that of the control group increased by an average of 0.5 cm leading to significant differences between groups (F = 4.395; *p* = 0.042). Finally, no significant differences were found between the waist circumference measured at 12 and 24 months (F = 1.199; *p* = 0.279).

For the hip circumference and waist–hip index, no significant differences were observed over time (*p* = 0.057 and *p* = 0.673, respectively) (Table 4). When comparing the groups, significant differences were found in the evolution of the hip perimeter among the participants of the experimental group (*p* = 0.001), but not among the participants of the control group (*p* = 0.927). Finally, there were no significant differences in the evolution of the waist–hip index between the two groups (control group: *p* = 0.875 and experimental group: *p* = 0.311) (Table 4).

### 3.4. Adherence to the Mediterranean Diet and Psychological Well-Being

Table 5 shows there was no significant change in the total scores of PREDIMED and GAD-7 questionnaires over time (time F = 1.399; *p* = 0.254 and F = 0.402; *p* = 0.671, respectively). Regarding the PHQ-9 questionnaire, a significant change over time was observed (time F = 4.376; *p* = 0.028). However, no significant differences were found when comparing between the experimental and control groups for these three questionnaires (PREDIMED: group × time F= 0.545; *p* = 0.582; PHQ-9: F = 0.504; *p* = 0.557; GAD-7: F = 1.482; *p* = 0.236, respectively) (Table 5). In the experimental group, the total score for the PREDIMED questionnaire was 9.2 at the beginning, 9.5 after 12 months of intervention, and 10.2 at the end of the intervention; the values for the control group were 7.8 at baseline and 8.0 and 8.4 after 12 and 24 months of intervention. No significant differences were found between groups when comparing the scores at baseline and the scores at 12 months (F = 0.033; *p* = 0.858), neither when comparing between baseline and after 24 months scores (F = 0.870; *p* = 0.357), nor between the scores at 12 and 24 months (F = 0.698; *p* = 0.409).

Referring to the PHQ-9 and GAD-7 questionnaires, the scores for the experimental groups were 3.5 and 3.9 at baseline, 1.8 and 0.9 after 12 months, and 1.1 and 1.0 after 24 months, respectively. For the control group, the scores for the PHQ-9 and GAD-7 were 4.5 and 5.2 at baseline, 1.8 and 2.1 after 12 months, and 1.0 and 1.1 after 24 months, respectively (Table 5). Similarly to the PREDIMED, no significant differences were found between groups when comparing between the three different times (PHQ-9: Baseline–12 months: F = 0.389; *p* = 0.538; Baseline–24 months: F = 0.693; *p* = 0.413; 12 months–24 months: F = 0.217; *p* = 0.645 and GAD-7: Baseline–12 months: F = 0.533; *p* = 0.471; Baseline–24 months: F = 2.626; *p* = 0.114; 12 months–24 months: F = 1.263; *p* = 0.270).

## 4. Discussion

The results of the present study indicate that after a two-year dietary nutritional intervention, the experimental group showed significant weight loss and percentage of fat mass, as well as a greater reduction in BMI and waist circumference. However, no differences were found between the groups in terms of hip circumference or waist–hip index. In relation to the PREDIMED, PHQ-9, and GAD-7 questionnaires, the results indicate that during these two years, the experimental and control groups improved their scores on the PHQ-9 questionnaire; however, no significant differences were observed between groups. Finally, there was no significant improvement in the PREDIMED and GAD-7 scores over time.

These results are similar to those obtained in other studies; Salas-Salvadó et al. [33] found significant differences in weight between the experimental group and the control group after 12 months of intervention (experimental group: −3.2 kg and control group: −0.7 kg; *p* < 0.001). In addition, when compared with the control group, they observed a greater reduction in waist circumference among the participants in the experimental group (experimental group: −3.3 cm and control group: −.7 cm; *p* < 0.001). Similarly, König et al. [34] found that the intervention program helped to significantly reduce body weight (pre: 102.5 kg and post: 96.3 kg; *p* < 0.001), BMI (pre: 35.6 kg/m^2^ and post: 33.5 kg/m^2^; *p* < 0.001), and waist circumference (pre: 113.1 cm and post: 106.3 cm; *p* < 0.001). In another study, a low-calorie MD pattern was claimed to be effective in reducing body weight, BMI, and fat mass; body weight decreased by 1.77 kg and BMI and fat mass decreased by 0.67 kg/m^2^ and 1.46 kg, respectively [35].

In relation to adherence to the MD, according to the results obtained, although no significant improvement was observed in either group, the experimental group increased their score from an initial 9.2 to a 10.2 at the end of the intervention. These findings are in line with those observed in other studies, such as Zuniga et al. [36], where a slight improvement was observed for the MD at the end of the intervention (experimental group: pre: 7.1, post: 8.7; control group: pre: 7.4, post: 7.6). A possible explanation for this slight improvement observed in our study could be the fact that our intervention coincided with the period of COVID-19 confinement, which could have made it harder for patients to adhere to a healthy dietary pattern and obtain better results in the PREDIMED questionnaire [37,38]. In addition, it must be taken into account that our sample already had good adherence to MD at the start of the intervention with very good values and, as described previously, it is the patients with baseline eating habits that are the farthest from a healthy diet who they are more likely to achieve greater changes in adherence [39].

The MD, owing to its foods and consumption frequencies, stands out as a potential means to help delay the progression of cognitive decline. Although no firm conclusions can be drawn at this moment, this dietary pattern is also considered a potential mean to reduce the prevalence of mental illness such as depression and anxiety [40,41]. In studies by Sánchez-Villegas et al. [20] and Mantzorou et al. [21], the MD pattern was associated with a lower risk of depression. In addition, in other studies such as the Sanchez-Villegas et al. [42] trial performed on older subjects, it was revealed that higher levels of brain-derived neurotrophic factor (BDNF), rendered by MD enhanced with walnuts and almonds, could prevent depression, memory loss, and cognitive decline. However, a recent study showed that the consumption of walnuts altered some metabolic syndrome biomarkers but did not significantly change the waist circumference, body weight, or BMI in older populations, stressing that the components of MD work in synergism to lower these markers in elders [43]. Similarly, Opie et al. showed that a wholesome MD rather than individual nutrients is recommended for the prevention of depression [44]. Jacka et al. [22], on his part, observed that a dietary pattern characterized by consumption of fruits, vegetables, beef, lamb, fish, and whole foods was associated with a lower risk of anxiety.

In the present study, it was observed that throughout the two-year intervention, both the experimental and control groups improved their scores on the PHQ-9 questionnaire; however, no improvement in the scores for the GAD-7 questionnaire was observed. A possible explanation for the lack of significant improvement in the scores of the GAD-7 questionnaire could be that, as mentioned previously, our intervention coincided with the period of COVID-19 lockdown, which generated a decrease in social interaction due to social distancing, which could have a negative impact on anxiety levels [45]. Nonetheless, this improvement in the scores of the PHQ-9 in both groups represents an important outcome for the improvement of people’s health. Especially, in older people, mental health is one of the most important aspects for health maintenance; mental health is important for their social participation, as in order to maintain their autonomy, older people must have optimal mental health [46]. In addition, psychological disorders such as depression and anxiety increase the risk of chronic diseases, including cardiovascular diseases and diabetes mellitus [47]. Therefore, the improvement of the participants in their score for the PHQ-9 questionnaire could generate, in turn, a reduction in the risk of chronic diseases mentioned previously and thus present a better state of health.

Thus, in general, our results suggest that a two-year intervention based on MD allows the older population with overweight or obesity to achieve greater weight loss, as well as a greater decrease in BMI, waist circumference, and percentage of fat mass.

This study has several limitations. First, the sample size was small, which may be associated with greater uncertainty regarding the measured effect. Because of the small sample size, the differences would have to be greater to reach statistical significance; therefore, although there is a real difference, its existence cannot be guaranteed. Second, this small sample size comes from a single city in Spain; therefore, caution should be used when generalizing the findings to other areas of Spain or Europe. Third, the measures were estimated based on self-reported data, which could imply that some participants underestimated their answers to the questionnaires.

## 5. Conclusions

A two-year intervention based on the MD pattern may be effective for weight and fat mass loss, as well as for the reduction of BMI and waist circumference in the older population with overweight or obesity. In relation to adherence to MD, no significant improvement was observed after two years of intervention. Finally, future research should explore the impact of the MD on mental disorders such as depression and anxiety.

## Figures and Tables

**Table 1 nutrients-14-04762-t001:** Sociodemographic and lifestyle variables. SD, standard deviation.

	Control Group	Experimental Group	All	*p*-Value
Sex Women Men	23 (88.5%) 3 (11.5%)	22 (88.0%) 3 (12.0%)	45 (88.2%) 6 (11.8%)	0.960
Mean age (SD)	66.58 ± 5.12	63.44 ± 3.50	65.04 ± 4.64	0.014
Marital status Married Divorced Widower	17 (65.4%) 3 (11.5%) 6 (23.1%)	19 (76.0%) 3 (12.0%) 3 (12.0%)	36 (70.6%) 6 (11.8%) 9 (17.6%)	0.328
Who are you living with? At home, alone/At my children’s home At home with my wife/husband or relatives	7 (26.9%) 19 (73.1%)	4 (16.0%) 21 (84.0%)	11 (21.6%) 40 (78.4%)	0.353
Level of studies No studies/Primary studies Mid-level studies Higher education	12 (46.1%) 10 (38.5%) 3 (11.5%)	5 (20.0%) 12 (48.0%) 8 (32.0%)	17 (33.3%) 22 (43.1%) 11 (21.6%)	0.021
Employment status Employed full or part time Unemployed or housewife Retired	5 (19.2%) 5 (19.2%) 16 (61.5%)	11 (44.0%) 5 (20.0%) 9 (36.0%)	16 (31.3%) 10 (19.6%) 25 (49.0%)	0.042
Tobacco use Yes, occasionally or on a regular basis No, I do not consume	2(7.7%) 23 (88.5%)	3 (12.0%) 21 (84.0%)	5 (9.8%) 45 (86.3%)	0.612
Alcohol consumption Yes, occasionally No, I do not consume	12 (46.5%) 13 (50.0%)	13 (52.0%) 12 (48.0%)	25 (49.0%) 25 (49.0%)	0.783

**Table 2 nutrients-14-04762-t002:** Nutritional status of the total simple at baseline. SD, standard deviation.

	Control Group	Experimental Group	*p*-Value
Mean weight (SD)	79.09 (11.87) kg	82.66 (13.99) kg	0.313
Mean BMI (SD) Overweight n (%) Obesity n (%)	32.17 (3.93) kg/m^2^ 9 (34.6%) 17 (65.4%)	32.70 (4.31) kg/m^2^ 5 (20.0%) 20 (80.0%)	0.826
Mean fat mass % (SD)	46.45 (5.24)	46.65 (6.17)	0.627
Mean waist-hip index (SD)	0.88 (0.06)	0.88 (0.05)	0.954
Cardiovascular risk Non-cardiovascular risk	17 (65.3%) 9 (34.6%)	15 (60.0%) 10 (40.0%)	

**Table 3 nutrients-14-04762-t003:** Changes in weight, BMI, and %FM in each of the groups.

							F (*p*-Value)	
	Baseline	12 Months	24 Months	Difference B-12M	Difference 12M-24M	Difference B-24M	Time	Group × Time
Weight (kg) Control group Experimental group	79.1 82.7	78.9 79.7	78.8 79.8	−0.2 −3.0	−0.1 +0.1	−0.3 −2.9	3.658 (0.039)	4.793 (0.017)
BMI (kg/m^2^) Control group Experimental group	32.2 32.7	32.1 31.5	32.1 31.6	−0.1 −1.2	±0 +0.1	−0.1 −1.1	3.701 (0.039)	5.020 (0.014)
Fat mass (%) Control group Experimental group	46.7 45.2	47.6 43.7	46.3 43.5	+0.9 −1.5	−1.3 −0.2	−0.3 −1.7	2.414 (0.098)	3.168 (0.049)

**Table 4 nutrients-14-04762-t004:** Changes in waist circumference, hip circumference, and waist–hip index.

							F (*p*-Value)	
	Baseline	12 Months	24 Months	Difference B-12M	Difference 12M-24M	Difference B-24M	Time	Group × Time
Waist circumference (cm) Control group Experimental group	100.1 101.0	101.6 98.7	100.6 98.5	+1.5 −2.3	−1.0 −0.2	+0.5 −2.5	1.430 (0.245)	5.248 (0.010)
Hip circumference (cm) Control group Experimental group	114.1 114.8	116.7 113.6	114.0 112.1	+2.6 −1.2	−2.7 −1.5	−0.1 −2.7	0.057	0.927 0.001
Waist-hip index (cm) Control group Experimental group	0.88 0.88	0.88 0.87	0.88 0.88	±0 ±0	±0 −0.1	±0 ±0	0.673	0.875 0.311

**Table 5 nutrients-14-04762-t005:** Changes in the scores of the PREDIMED, PHQ-9, and GAD-7 questionnaires.

							F (*p*-Value)	
	Baseline	12 Months	24 Months	Difference B-12M	Difference 12-24M	Difference B-24M	Time	Group × Time
PREDIMED (total score) Control group Experimental group	7.8 ± 1.5 9.2 ± 1.3	8.0 ± 2.3 9.5 ± 1.7	8.4 ± 2.0 10.2 ± 1.4	+0.2 +0.3	+0.4 +0.7	+0.6 +1	1.399 (0.254)	0.545 (0.582)
PHQ-9 (total score) Control group Experimental group	4.5 ± 4.6 3.5 ±3.3	1.8 ± 2.5 1.8 ± 2.5	1.0 ± 1.5 1.1 ± 2.0	−2.7 −1.7	−0.8 −0.7	−3.5 −2.4	4.376 (0.028)	0.504 (0.557)
GAD-7 (total score) Control group Experimental group	5.2 ± 4.0 3.9 ± 3.9	2.1 ± 5.2 0.9 ± 1.5	1.1 ± 2.3 1.0 ± 1.8	−3.1 −3.0	−1.0 +0.1	−4.1 −2.9	0.402 (0.671)	1.482 (0.236)

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
