# Peer review of "Effectiveness of a Two-Year Multicomponent Intervention for the Treatment of Overweight and Obesity in Older People"

_nutrients, 2022, doi:10.3390/nu14224762_

Round 1

Reviewer 1 Report

This study evaluated the effectiveness of a 2-year Mediterranean dietary nutritional intervention among older adults with overweight or obesity issues. The findings showed significant intervention effects on weight loss and mental well-being.

A few issues should be addressed:

1. The title says "dietary-nutritional intervention." However, the actual intervention included a physical activity component, and also psychological support and self-care recommendations. In other words, it seems like the intervention was more than just nutrition/dietary. The authors might consider revising their description/title of the intervention. This is particularly relevant because the experiment group did not show greater adherence to the Med diet. It was possible that the dietary component was not as effective as the other components in weight loss and mental health improvement.

2. Relevant to the first point: the authors should discuss further the lack of improvement in MD adherence in the experiment group. The authors discussed previous studies, and one possible explanation being the impact of the COVID-19 pandemic.  But what does it mean for the study on intervention efficacy? If the intervention wasn't improving MD adherence significantly more than the control group, then what is causing the significant effects on weight loss? This is a major point that is yet to be fully discussed.

3. More details are needed on the intervention components. How were the contents designed? What were the sources? Who delivered them? In what setting? Online or in person? How did the study document if all participants in the experiment group received the education?

Author Response

REVIEWER 1

  1. The title says "dietary-nutritional intervention." However, the actual intervention included a physical activity component, and also psychological support and self-care recommendations. In other words, it seems like the intervention was more than just nutrition/dietary. The authors might consider revising their description/title of the intervention. This is particularly relevant because the experiment group did not show greater adherence to the Med diet. It was possible that the dietary component was not as effective as the other components in weight loss and mental health improvement.

Thank you very much for your suggestion. We have modified the title of the manuscript taking into account everything you mentioned in your comment.

“Effectiveness of a Two-Year Multicomponent Intervention for the Treatment of Overweight and Obesity in Older People”

  1. Relevant to the first point: the authors should discuss further the lack of improvement in MD adherence in the experiment group. The authors discussed previous studies, and one possible explanation being the impact of the COVID-19 pandemic. But what does it mean for the study on intervention efficacy? If the intervention wasn't improving MD adherence significantly more than the control group, then what is causing the significant effects on weight loss? This is a major point that is yet to be fully discussed.

We have explained why we think MD adherence was not improving significantly more than the control group. We think that apart from COVID-19, another reason could be the fact that our sample already had a good adherence to the MD at the start of the intervention with very good values (9.0 score). In line with this, it has been previously described that the patients with baseline eating habits that differ most from a healthy diet are likely to achieve greater adherence changes.

  1. More details are needed on the intervention components. How were the contents designed? What were the sources? Who delivered them? In what setting? Online or in person? How did the study document if all participants in the experiment group received the education?

We would like to thank you for your comments and suggestions because they can help us to improve the quality of our manuscript. We have given more details related to all the topics you mentioned in your comment related to the intervention. The new information included is the one which is highlighted in green.

“The contents of the intervention were prepared and designed by the multidisciplinary research team: dietitians-nutritionists, doctors, psychology staff, nursing staff and physiotherapists. For the design, the research team, based on the scientific evidence in the literature and on the recommendations of official bodies such as the World Health Organization.

The multicomponent intervention was carried out in person in the “Health Class-room” of the University of Alicante. As described in Zaragoza-Martí et al., [23], partici-pants in the experimental group received personalized training for weight management, food education, and psychological support and self-care recommendations. The die-tary-nutritional intervention was carried out by a trained and qualified dieti-cian-nutritionist. During de individual sessions, a menu adapted to their needs based on the MD was provided to each participant. In addition, in the follow up individual ses-sions, the dietitian-nutritionist monitored weight loss by taking anthropometric meas-urements and evaluated eating behavior by passing questionnaires. During the group sessions, topics related to the MD and its health benefits, the nutritional labeling of packaged foods, the key diet for healthy aging, seasonal diet, the possibility of a healthy gastronomy, diet and cholesterol, diet and hypertension prevention, diet and bone health, healthy snacks and healthy recipes were carried out. In the psychological support sessions, issues related to self-control, anxiety management, emotional feeding, achievements, and difficulties encountered during treatment, review of weekly action plans, relaxation techniques, consolidation of the new image, cognitive distortions, change management, coping with uncertainty, and motivation to change were discussed. The psychological follow-up was carried out by the psychology staff and the follow-up of self-care in health was carried out by the nursing staff. For physical activity, following the recommendations of the World Health Organization (WHO), the practice of at least 150 minutes per week was recommended.”

Reviewer 2 Report

This Mediterranean diet intervention two-year study in an older population showed lower markers for weight, waist circumference, BMI, and fat mass percentage, while depression levels improved

The subject is important and the manuscript is well writtenthe abstract includes sufficient informationthe introduction summarizes the topic current state and knowledge in the fieldthe methods are described with sufficient details, the results are accurately presented, with relevant data given in tables. In the discussion section, including the following suggestions, the paper could be improved:

A trial performed on older subjects revealed that higher levels of brain-derived neurotrophic factor (BDNF), rendered by MD enhanced with walnuts and almonds, could prevent depression, memory loss, and cognitive decline (doi: 10.1179/1476830511Y.0000000011). However, a recent study showed that consumption of walnuts altered some metabolic syndrome biomarkers but did not significantly change waist circumference, body weight, or BMI in older populations stressing that the components of MD work in synergism to lower these markers in elders (doi: 10.3390/antiox11071412). Similarly, Opie et al. showed that wholesome MD rather than individual nutrients is recommended in the prevention of depression (doi: 10.1179/1476830515Y.0000000043).

Author Response

REVIEWER 2

In the discussion section, including the following suggestions, the paper could be improved: A trial performed on older subjects revealed that higher levels of brain-derived neurotrophic factor (BDNF), rendered by MD enhanced with walnuts and almonds, could prevent depression, memory loss, and cognitive decline (doi: 10.1179/1476830511Y.0000000011). However, a recent study showed that consumption of walnuts altered some metabolic syndrome biomarkers but did not significantly change waist circumference, body weight, or BMI in older populations stressing that the components of MD work in synergism to lower these markers in elders (doi: 10.3390/antiox11071412). Similarly, Opie et al. showed that wholesome MD rather than individual nutrients is recommended in the prevention of depression (doi: 10.1179/1476830515Y.0000000043).

We would like to thank you for your comments and suggestions because they can help us to improve the quality of our manuscript. We have included the suggestions you have made in your comment in our discussion section the following way:

“The MD, owing to its foods and consumption frequencies, stands out as a potential means to help delay the progression of cognitive decline. Although no firm conclusions can be drawn at this moment, this dietary pattern is also considered a potential mean to reduce the prevalence of mental illness such as depression and anxiety [40,41]. In studies by Sánchez-Villegas et al. [20] and Mantzorou et al. [21], the MD pattern was associated with a lower risk of depression. In addition, in other studies such as the Sanchez-Villegas et al [42] trial performed on older subjects  revealed that higher levels of brain-derived neurotrophic factor (BDNF), rendered by MD enhanced with walnuts and almonds, could prevent depression, memory loss, and cognitive decline. However, a recent study showed that consumption of walnuts altered some metabolic syndrome biomarkers but did not significantly change waist circumference, body weight, or BMI in older populations stressing that the components of MD work in synergism to lower these markers in elders [43]. Similarly, Opie et al. showed that wholesome MD rather than individual nutrients is recommended in the prevention of depression [44]. Jacka et al. [22] on his part observed that a dietary pattern characterized by consumption of fruits, vegetables, beef, lamb, fish, and whole foods was associated with a lower risk of anxiety.”
